# Computer Life-Cycle Management System for Avionics Software as a Tool for Supporting the Sustainable Development of Air Transport

Mariusz Zieja , Andrzej Szelmanowski, Andrzej Pazur and Grzegorz Kowalczyk *

Logistic Support Department, Air Force Institute of Technology, 01-495 Warsaw, Poland;
mariusz.zieja@itwl.pl (M.Z.); andrzej.szelmanowski@itwl.pl (A.S.); andrzej.pazur@itwl.pl (A.P.)
* Correspondence: grzegorz.kowalczyk@itwl.pl; Tel.: +48-261-851-310

**Abstract:** The article presents selected results of analytical and design works undertaken at the Air Force Institute of Technology (AFIT) in the field of building a computer support and software lifecycle management system that is critical for flight safety. The aim of the work undertaken is to develop methods and carry out verification and testing in order to detect errors in the developed avionics software for compliance with the requirements of the DO-178C standard and its production, certification, and implementation on board aircraft. The authors developed an original computer system within the implemented requirements used in the construction and certification of avionic onboard devices and their software (among others, DO-254, DO-178C, AQAP 2210, ARP 4761, ARP 4754A). The conducted analysis involved three basic groups of avionics software development processes, i.e., software planning, creation, and integration. Examples of solutions implemented in the constructed computer system were presented for each of these process groups. The theoretical basis of the new method for predicting vulnerabilities in the software implemented within integrated avionic systems using branching processes is discussed. It was demonstrated that the possibility of predicting vulnerabilities in future software versions could have a significant impact on assessing the risk associated with software safety in the course of its lifecycle. It was indicated that some of the existing quantitative models for analyzing software vulnerabilities were developed based on dedicated software data, which is why actual scenario implementation may be limited. DO-178C standard requirements for the process of developing avionics software were implemented in the helmet-mounted flight parameter display system constructed at AFIT. The requirements of the DO-178C and AQAP 2210 standards were shown to be met in the example of the software developed for a graphics computer, managing the operating modes of this system.

**Keywords:** sustainable air transport; computer system; software security; DO-178C standard; AQAP 2210 standard; branching process; helmet-mounted flight parameter display system

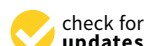

## 1. Introduction

Contemporary aircraft, both civilian and military, are equipped with various radio–electronic onboard devices, which support a pilot in executing complex air missions. The rapid development of digital electronics and IT, increasingly distinct over the recent years, requires a pilot of a modern aircraft to have the in-flight support of onboard equipment, which, in fact, are already "smart" computers with sophisticated and comprehensive avionics software. Such computerization of a modern aircraft should ensure its required operating reliability and the safety of both the crew and the passengers [1–4].

The process of producing digital onboard devices and systems enables their reliable operation under all flight conditions that are critical to them, among others, due to changes in the pressure and temperature of ambient air and the occurrence of linear overloads, e.g., during a maneuvering flight. It is quite easy to imagine the consequences of, e.g.,

ILS landing-system software failure during an aircraft's approach to an airport in difficult weather conditions or damage to a helmet-mounted weaponry control system when indicating a target, selecting weapon type, or using it in the course of a combat flight.

In order to satisfy the stringent reliability-related requirements within the process of developing electronic air equipment and their dedicated avionics software, specialists have drawn up relevant standards, such as standard DO-178C, containing software requirements, and standard DO-254, containing hardware requirements, supported by additional standardizing documents, including ARP 4761 and ARP 4754A (Figure 1).

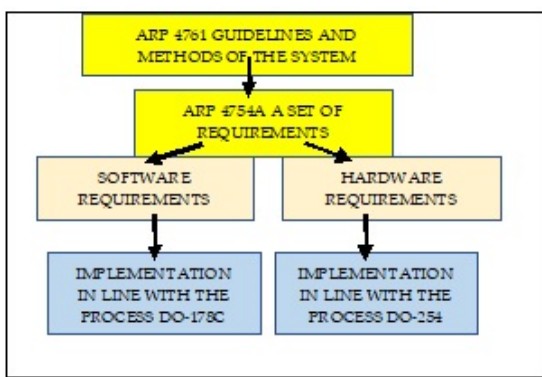

**Figure 1.** Manufacturing process diagram for avionic devices and software critical to flight safety, as per standard DO-178C.

Standard ARP 4761 is a set of processes and techniques used to ensure the required assurance level of designed devices and systems; it is intended for installation on board aircraft. This standard defines five damage levels for onboard equipment and their consequences [5]. Level A covers damage, the occurrence of which may lead to severe casualties and material losses (e.g., air crash). Level B covers major damage that may lead to severe injuries (e.g., injury of a pilot or passengers). Level C covers major damage of a lower exposure degree, meaning the failure of main onboard systems but the flight being allowed to continue (e.g., through manual steering). Level D covers minor damage, e.g., when the flight is allowed to continue without a specific functionality (e.g., main radio damage). Level E covers damage, the occurrence of which is insignificant to the flight continuation process and pilot's actions (e.g., mechanical wear of steering element buttons). Standard ARP 4754A is a set of functional system requirements in terms of software and hardware.

Standard DO-178C is the basic document applied within the international aviation market to the process of manufacturing and certifying the software of onboard electronic devices. It consists of a basic part and supplementary standards DO-331, DO-332, and DO-333 as well as DO-330, DO-248C, and DO-278A, which define additional requirements and activities to be provided for in the software development process, depending on the adopted implementation technique (Figure 2).

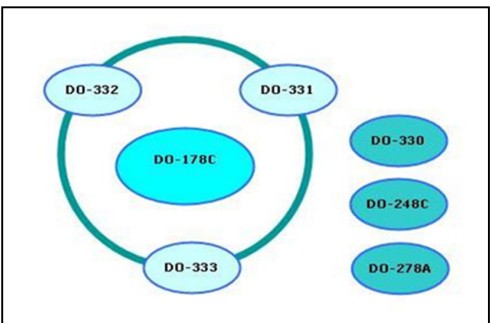

**Figure 2.** Relationships between supplementary standards that define additional requirements and activities under standard DO-178C.

Standard DO-331 describes software and hardware designs using a modeling technique (model-based designs). Computational algorithms or their fragments are implemented using external tools, and the designing process is executed by combining blocks that execute various mathematical operations. A computation diagram implemented this way is subject to testing and verification. The final stage of algorithm implementation is the automatic generation of a diagram based on the developed software source code. When using this method, computational algorithms are implemented visually, with automatic generation of the resultant software code.

Standard DO-332 is applied when projects are implemented with the use of object-oriented programming languages (including C++, Java, Ada). The fundamental issue in the case of object-oriented languages is the verification of such software, which follows three basic techniques, namely, inheritance, polymorphism, and dynamic linking. An embodiment of such a standard can be the occurrence of yet another method of the same name within the same class hierarchy, which ensures that when a relevant class is called, the appropriate method is executed.

Standard DO-333 is applied when formal methods are used within the designing process. Formal methods are mathematical techniques, which are applied within the software development and verification process. The task of formal methods is to develop a mathematical system for the constructed system and to verify its behavior in order to prove that the system under construction is functional and satisfies the assumed safety requirements. Standard DO-333 describes the qualification process for software tools used within the process of software development, testing, launching, and modification. A qualification process is understood as an evaluation of software tools, the output of which is not verified and which simplify, accelerate, and automate a DO-178C standard process.

Standard DO-248C is an additional and supplementary document for the DO-178C standard, explaining its incomprehensible and problematic aspects.

The above requirements of the DO-178C standard became the basis for the ITWL to build a computer system supporting the lifecycle management of avionic software, developed and certified for electronic devices, where the main goal is to develop software and implement it into the SWPL-1 CYKLOP helmet flight-parameter display system [6–8].

## 2. Materials and Methods

The lifecycle of avionics software is associated with the implementation of numerous, various types of interlinked tasks, the objective of which is to design and develop software of appropriate quality that meets the requirements of the ordering party as well as to ensure the required security level of the designed avionic systems and devices [9–11].

Standard DO-178C defines three primary projects, namely, software planning, development, and integration, which are mutually concurrent within selected areas. This means that the commencement of one process does not have to be linked with the completion of a previous one. An example would be starting the tests of an encoded software fragment prior to the stage of total encoding completion [12–17].

There are many software lifecycle models, which include the cascade model. The following major steps, to be executed in order to meet the requirements of standard DO-178C, are distinguished within this model. They involve collecting the requirements and their analyses as well as software design, development, testing, and implementation.

Each of these steps allows us to go back to the previous state, enabling interaction between the executed tasks, which is of particular importance in the case of detecting software bugs that have to be fixed. The implementation of the developed avionics software also covers its operation until its disposal.

### 2.1. Avionics Software Planning Process and Possible Computer-Aided Support

Software planning involves processes associated with planning and standards, after which software development will follow. At this stage, standard DO-178C defines five

major plans and three software standards. It also distinguishes other documents that are required for the certification of any developed software [18–20].

The major avionics software plans are as follows:

- Plan for Software Aspects of Certification (PSAC): a kind of "contract" between the contractor and the certification body.
- Software Development Plan (SDP): contains the requirements concerning software planning, coding, and integration stages. The SDP was written for software developers and is, a kind of a guide on how to develop the software in order for it to satisfy the adopted requirements.
- Software Verification Plan (SVP): contains aspects associated with verifying software functionality and is intended for software testers. The SVP is associated with the SDP because the assumptions that were thought out at the software development stage are verified at this stage.
- Software Configuration Management Plan (SCMP): defines procedures, tools, and methods aimed at achieving the objectives associated with managing the requirements throughout the entire software lifecycle. It covers the procedures in terms of defining the baseline version and identifying software versions, reporting issues, controlling and reviewing modifications, archiving, controlling software loading and recovery.
- Software Quality Assurance Plan (SQAP): defines the procedures and methods to be applied in order to satisfy the quality requirements associated with standard DO-178C. It determines the procedures in terms of quality management, audit execution, actions associated with issue reporting, and corrective action methodology.

The major software development standards are as follows:

- Software Requirements Standards (SRS), which define principles, methods, and tools to be applied for developing high-level requirements. They include methods used for software development, notations for requirement implementation (algorithms, flow diagrams), and project tool limitations, which will be used for software development, as well as criteria related to the requirements.
- Software Design Standards (SDSs): They define methods, tools, and limitations within the software design process. They include low-level requirements and software architecture. They are intended for a software development team and explain how to implement a design effectively. They cover, among other things, nomenclature methods, design tool limitations, and software complexity (e.g., procedure length).
- Software Coding Standards (SCSs); They define methods, tools, and limitations within the software coding process. They include, among other things, coding standards, programming languages used, code presentation standards, nomenclature standards, compilator limitations, and restrictions arising from programming standards.
- The aforementioned documents, as reference templates, have been implemented within the constructed computer system, which supports the process of managing avionics software development and certification.

### 2.2. Avionics Software Development Process and Possible Computer-Aided Support

The avionics software development process consists of four detailed processes:

- The software requirement process, which leads to the development of high-level requirements (HLRs);
- The software design process, which develops low-level requirements (LLRs) and software architecture based on HLRs;
- The coding process, which leads to the creation of a source code and a nonintegrated object code;
- The software integration process, which involves consolidating the software into the form of executable programs and its integration with external devices.

High-level requirements (HLRs) are implemented based on system architecture and system requirements. They involve time waveforms, memory management, planned

links with external devices, methods of responding and detecting errors, system operation monitoring, and software partitioning. HLRs constitute a base to develop low-level requirements used in the software design process, which include descriptions of connections with external devices, definitions and manner of data flow, communication mechanisms, and software components.

The coding process involves translating LLRs into source code and a precompiled object code. It is associated with the verification process because the preliminary execution of a partially developed code is conducted at this stage. The integration process involves compiling and combining the compiled code into executable applications (one or more) and embedding this software on the target platform (the onboard device).

Software development processes determine one or many system requirement levels. High-level requirements are determined based directly on system architecture and system requirements. They are developed within the software design process, thus creating interrelated low-level requirements. However, when a source code is generated based directly on high-level requirements, it also corresponds to low-level requirements and is subject to recommendations on low-level requirements. Software requirement processes utilize software lifecycle process outputs to create high-level requirements. The basic result of this process is software requirement data.

Software requirement data define high-level requirements, including the requirements provided by the ordering party. The data should include a description of the software system requirement allocation, taking into account the safety requirements and potential error conditions, functional and operational requirements for each operating mode, performance criteria (e.g., precision and accuracy), time-related requirements and limitations, memory size limitations, hardware and software interfaces (e.g., protocols, formats, input/output frequency), error detection, safety monitoring, as well as the requirements of software partitioning (how separated software components cooperate) and software levels for each component.

The input into the avionics software design process includes software requirements, a software development plan, and software design standards. If the planned transition criteria are met, the high-level requirements are used within the manufacturing process for creating software architecture and low-level requirements. They may include one or more requirement levels.

The basic output in these processes is the design description, which contains software architecture and low-level requirements. The data should include a detailed description of how the software satisfies high-level requirements, including algorithms and data structure, and how the software requirements correspond to the processes and tasks. It should also provide the software architecture description defining software structure, with implemented requirements, input/output descriptions (e.g., data dictionary, data and control flow within the design), resource limitations, a management strategy for resources and their limitations, and margins as well as methods for measuring these margins (e.g., time and memory, sequencing procedures). The description should include intraprocessors and intratask communication mechanisms, including fixed interruption time sequences, design methods, and details on their implementation (e.g., software loading). An important element of the description is user-modified software, partitioning methods, and measures to prevent partition breach as well a description of software components (regardless of whether they are new or previously manufactured) and a reference to the baseline version from which they were downloaded. This description should also include derivative requirements arising from the software design process. If a system contains an inactive code, a description of security measures for the activation of the code on the target computer and a justification of design decisions are directly included in the system requirements associated with its security.

The software design process is complete when its objectives and the goals of the associated integration processes are achieved. Within the software coding process, the source code is implemented based on software architecture and low-level requirements. Coding

process inputs are low-level requirements and software architecture from the software design processes, the software development plan, and software coding standards. The software coding process can be commenced when planned transition criteria are met. The source code is developed in the course of this process and is based on system architecture and low-level requirements. The target computer and the source code from software coding processes are used in order to compile, combine, and load data within the integration process; this is aimed at integrating an avionics system or its equipment constituents.

### 2.3. Avionics Software Integration Process and Possible Computer-Aided Support

The avionics software integration process consists of four detailed processes:

- Communication within software certification;
- Requirement management within software certification;
- Verification within software certification;
- Quality assessment within software certification.

Communication within certification is an essential process, the task of which is the successful completion of software certification. It involves constant cooperation and communication between the applicant and the certification body. The applicant is the entity seeking certification. This process spreads over the entire software lifecycle, which starts with planning and ends with its disposal. The task of the applicant is to determine compliance measures, which define the manner in which the software will satisfy basic certification requirements.

The process of requirement management, just like communication within certification, spreads over the entire software lifecycle. It covers all data and documentation used for software development and verification. Managing the requirements is the "art" of identifying, organizing, and controlling changes at the software development stage. The main task of this process is to achieve the highest possible efficiency while minimizing errors. The requirement management method is associated with the onboard equipment damage level. Standard DO-178C defines two software control levels: Levels CC1 and CC2. Level CC1 must satisfy all DO-178C requirements, whereas Level CC2 only satisfies some of them (related to Assurance Levels C and D).

The software verification process involves the detection and description of errors introduced from the software planning stage to the development stage. Standard DO-178C does not define the techniques utilized for verification but rather the objectives that must be achieved.

The task of the quality assessment process is to demonstrate that the developed software is compliant with the assumed requirements and standards, which, in consequence, should result in a product meeting the expectations of the ordering party (or show discrepancies relative to the adopted requirements). Software quality assessment is an ongoing process, which starts at the planning stage and continues through the development and testing stages until the final product.

All of the aforementioned features can be satisfied with regards to developed avionics software through the use of appropriate computer-aided management [21].

### 2.4. A Method for Predicting Avionics Software Vulnerabilities Using Branching Processes

One of the latest approaches aimed at dealing with avionics software security vulnerabilities is the application of vulnerability and hacker-attack predicting models (VPMs). These models are based on machine-learning elements, which enables predicting software components that may contain vulnerabilities in their future versions. VPM modules utilize software attributes from their historical versions as input data, which are then used in binary classification [22,23].

The most common learning techniques used in modeling software vulnerabilities are, e.g., logistic regression, decision trees, k-nearest neighbors, naive Bayes, random forest and support vector machine (SVM).

The conducted analysis indicated the two most popular VPM types:

- The use of software metrics, which take into account a specific set of software metrics when creating a binary classifier. The objective of preliminary testing is the empirical evaluation and confirmation of expert opinions that software complexity is opposed to software security. However, the generally observed weak link between complexity and security vulnerabilities lead to the need to investigate various models for predicting vulnerabilities, such as code modification, relationship, code coverage, conjugation, consistency, and developer activity.

- The use of text exploration techniques, where the source code of tested software components is parsed and represented as a set of tokens (i.e., keywords). Tokens are combined into a data set and user together with the data on vulnerabilities for training vulnerability predictors. During the second phase (called the prediction phase), a trained classifier uses these datasets to determine whether a future version of a studied code module is vulnerable to errors and hacker attacks or not.

The starting point of the developed method is determined according to which software security can be defined as follows [24,25]:

$$R(t) = 1 - Q(t) \tag{1}$$

where $R(t)$ is the software security state (no vulnerabilities discovered); $Q(t)$ is the vulnerable state (vulnerabilities discovered in the examined software); $\lambda(t)$ is the intensity of transition from state $R(t)$ to $Q(t)$.

Software vulnerability occurrence intensity is defined as the density of vulnerability occurrence probability in time, provided that no software vulnerabilities are discovered during that time [26–30].

Software vulnerability occurrence intensity is described by the following formula:

$$\lambda(t) = \lim_{\Delta t \to 0} \frac{P\{t < T \le t + \Delta t | t < T\}}{\Delta t} \tag{2}$$

The probability of discovering vulnerabilities in a single software occurrence is expressed by the formula:

$$q = 1 - e^{-\lambda t} \tag{3}$$

where $q$ is the probability of vulnerability discovery in a single software instance; $t$ is the time of a single software instance operation in time interval $(0, t)$.

Further transformations allowed to determine estimator ($\hat{\lambda}$) of the unknown intensity of discovered vulnerabilities—parameter $\lambda$. For this purpose, the maximum likelihood method was used, which gives the result that can be written as follows:

$$\hat{\lambda} = \frac{n_1 + n_2 + \ldots + n_i}{T_1 + T_2 + \ldots + T_i} \tag{4}$$

The proposed method involves calculating the probability of eliminating software vulnerabilities and the mean value of eliminated vulnerabilities (at a relevant time). The proposed approach is based on using branching processes [31].

The method concept involves patching vulnerable software. Firstly, a scenario with applied antivulnerability measures is considered. In such a case, it is assumed that after one testing period, all software vulnerabilities are detected and patched. As a result, these former vulnerabilities are either nonexistent in the following period or their impact falls to a safe level; however, new vulnerabilities appear due to patching the previous ones. Furthermore, there is always at least one software vulnerability, so for the sake of simplifying the notation, it is assumed that the zero period has at least one vulnerability. Let it denote the number of sensitivity descendants for each of them. This means that a new population size appears from time to time, which can be obtained as follows:

$$Q(n + 1) = \sum_{k=Q(1)}^{Q(n)} Y_k^{(n+1)} \tag{5}$$

where $(Y_k)_{k \geq 1}$ forms a sequence of independent, identically distributed, non-negative (and almost surely finite) random variables. Note that in this model, the $Q(n)$ vulnerabilities in the $n - th$ period of time are patched at time $n + 1$ as they are not considered in (11).

Let $G_n(s) = E\left[s^{Q(n)} \middle| Q(0) = 1\right]$ now denote the probability-generating function of $Q(n)$, $n \in N$ ($N$ is a set of non-negative integers) with $G_0(s) = s$ and $G_n(0) = P(Q(n) = 0 | Q(0) = 1)$, where $E[\cdot | \cdot]$ denotes a conditional expectation. Moreover, let us assume that $\mu = G_1'(1)$, where by $G_1'$ we mean the derivative of $G_1$ (let us also recall that the expression '$f : g$' means $f$ is equal to $g$ by definition).

Let $\mu_n$ denote the mean population size at generation n $\geq 0$. We have the following well-known fact:

**Theorem 1.**

$$\mu_n = E[Q(n) | Q(0) = 1] = (E[Q(1) | Q(0) = 1])^n = \mu^n, \qquad n \geq 1 \tag{6}$$

From the above theorem, it is easy to see that

$$\begin{aligned} E[Q(n) | Q(0) = k] &= kE[Q(n) | Q(0) = 1]^n \\ &= k(E[Q(1) | Q(0) = 1]^n = k\mu^n \end{aligned} \tag{7}$$

The time of vulnerability elimination $T_0$ can be defined by

$$T_0 = \inf\{n \geq 0 | Q(n) = 0\} \tag{8}$$

where "inf" is the infimum of a set.

Let us denote the probability of elimination within a finite time interval, starting from $Q(0) = k$ by $\alpha_k$, i.e.,

$$\alpha_k := P(T_0 < \infty | Q(0) = k) \tag{9}$$

Similarly, as in Theorem 1, we can check that

$$\begin{aligned} \alpha_k &:= P(T_0 < \infty | Q(0) = k) = (P(T_0 < \infty | Q(0) = 1))^k \\ &= \alpha_1^k, \ k \geq 1 \end{aligned} \tag{10}$$

The above values can be computed using the following theorem:

**Theorem 2.** *The probability of vulnerability elimination* $\alpha_1 := P(T_0 < \infty | Q(0) = 1)$ *is the smallest solution of the equation* $G_1(s) = s$.

Now, let us assume that every generation of vulnerability is given by the Poisson distribution; in particular, we have $P(Y_1 = n) = \frac{\hat{\lambda}^n}{n!} e^{-\hat{\lambda}}$, $\hat{\lambda} > 0$, where $\hat{\lambda}$ is estimated from (4).

Under the assumption that $Q(0) = 1$, one gets

$$\begin{aligned} G_1(s) &= E[s^{Y_1}] = \sum_{n=1}^{\infty} s^n P(Y_1 \, n) \\ &= e^{-\hat{\lambda}} \sum_{n=0}^{\infty} s^n \frac{\hat{\lambda}^n}{n!} = e^{\hat{\lambda}s - \hat{\lambda}} \end{aligned} \tag{11}$$

Moreover, $G_1'(s) = \lambda e^{\hat{\lambda}s - \hat{\lambda}}$; therefore, by using Theorem 1, we obtain

$$E[Q(n) | Q(0) = 1] = \mu^n = (G_1'(1))^n = \hat{\lambda}^n \tag{12}$$

Assuming that $Q(0) = k$, $E[Q(n) | Q(0) = k] = k\hat{\lambda}^n$.

By Theorem 2, the probability of eliminating vulnerabilities (assuming that Q(0) = 1) is the smallest solution to the following equation:

$$e^{\hat{\lambda}s-\hat{\lambda}} = s \tag{13}$$

Assuming that $s \in [-1, 1]$ and $e^x > 0$ for any $x \in R$, where R is a set of real numbers if we restrict to seek solutions within the interval $(0, 1]$, obviously 1 is the solution of this equation. Under the next assumption that Q(0) = k from (10), it is enough to raise (13) to a power k.

We will next show that for $\hat{\lambda} \in (0, 1)$, the smallest solution of (13) is 1. In other words, it is necessary to show that for any $s \in [0, 1]$, the tangent line to the graph of $f(s) = e^{\hat{\lambda}s-\hat{\lambda}}$ has a strictly smaller slope than the function $g(s) = s$.

Indeed, for $\hat{\lambda} \in (0, 1)$, we have

$$f'(s) = \hat{\lambda}e^{(\hat{\lambda}s-\hat{\lambda})} < e^{(\hat{\lambda}s-\hat{\lambda})} < e^{(\hat{\lambda}s)} < 1 = g'(s) \tag{14}$$

Using the same method, we can show that for $\hat{\lambda} = 1$, Equation (13) has exactly one solution, i.e., s = 1. Let us then consider function $F(\hat{\lambda}) = e^{(\hat{\lambda}s-\hat{\lambda})}$. It is easy to see that if we set s = 1, then for any $\hat{\lambda}$, this function is equal to 1. Hence, F is strictly decreasing, and, by (14), we obtain that for $\hat{\lambda} \in (1, +\infty)$, there is always $s \in [0, 1)$, which is the solution of (13).

In our case, this means that when setting $\hat{\lambda} \leq 1$, then the probability of eliminating vulnerabilities will be equal to 1, and the mean value of vulnerabilities will converge to 0 by (12). On the other hand, if we set $\hat{\lambda} > 1$, the probability of eliminating vulnerabilities will always be smaller than 1, and the mean value of vulnerabilities will diverge to $+\infty$.

As we were saying, the most interesting situation is when the probability of eliminating vulnerabilities is equal to 1. Otherwise, when the probability is less than 1, we are not sure of elimination.

Finding avionics software security vulnerabilities that lead to errors or hacker attacks is considered a useful activity that can highly impact flight safety and reliability. The ability to predict the occurrence of software vulnerabilities or to quantitatively measure their impact enables the forecasting of software security trends and the planning of a widely understood process of managing its safety.

The developed method is aimed at improving the ability to predict vulnerabilities in the tested software. Verifying and then improving the accuracy of the proposed method requires further research, followed by empirical analysis using the data from vulnerability databases or other types of resources on vulnerability.

## 3. Results

One of the solutions introduced at AFIT in the field of limiting errors within developed avionics software is a computer-aided management system, as per the requirements of standard DO-178C, and the implementation of these requirements in the form of a procedure in the ISO-9001 Quality Assurance System. The constructed computer system enables the implementation of verifications and the creation of documents required by standard DO-178C (i.e., plans, reports, statements) directly from the AFIT IT network.

### 3.1. Basic Tasks and Functions of a Computer System Supporting the Management Process

The task adopted for the purposes of implementation was constructing a tool for computer-aided management of software development for the purposes of a helmet-mounted flight-parameter display system, to be developed and constructed at AFIT as per standard DO-178C.

The main functions of the constructed computer support IT system included:

- Setting up a new project, which involves entering information on, among other things, project title, personal data of the project manager and individual contractors, their authorizations, and system accessibility levels to the system;

- Entering data into the knowledge base regarding project implementation (details, finances, and limitations);
- Automatic generation of document templates required in standard DO-178C (i.e., plans, standards, verification procedures and methods, reports, and other entries);
- Automatic generation of tests for the developed software and archiving the test results;
- Archiving correspondence between project contractors, program files, and their test results;
- Automatic backup of the files to a server located in another building within AFIT premises (protection against data loss);
- Providing project implementation data as per entered authorization of system users;
- Reporting project status for the purposes of an audit or inspection, as per the entered guidelines.

A computer system supporting the management of avionics software management, after installing specialized software (including static and dynamic analyses, adapted to software vulnerability and error detection, to determine its vulnerabilities to structural damage of the avionics system and hacker attacks), enables direct, electronic cooperation between project participants and its supervision by the project manager, who is responsible for correct project implementation.

### 3.2. Structural Diagram of a Computer System Supporting the Management Process

The main structural element of the computer system supporting the management of avionics software development is a special server built into the AFIT IT network (Figure 3). The server cooperates with the workstations of individual system users. A file server, called a backup, is used to protect the gathered information. Its task is to archive the files. The current state of project progress is saved in its memory after each "working day". It also cooperates with an emergency server, which is turned on in the case of main server failure (protecting the current project's progress).

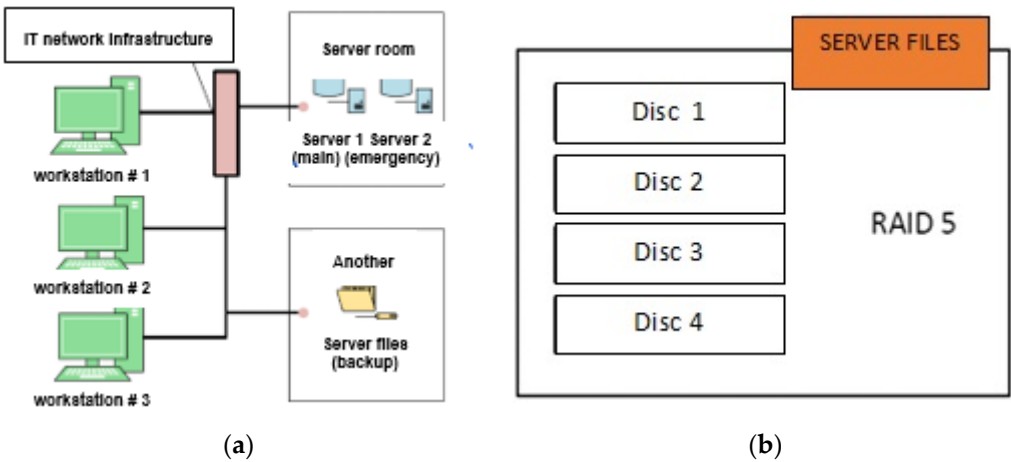

|                 |                 |
| :---: | :---: |
| (**a**)         | (**b**)         |

**Figure 3.** Structure of a computer system supporting the development of avionics software critical to flight safety; (**a**) Architecture diagram for a computer system integrated with an IT network; (**b**) Architecture diagram of a file server.

The main server operating software includes a Windows Server operating system, a Windows SQL Server database, and an MS Office editing and calculation suite as specialized software for project management using guidelines as per standard DO-178C.

The specialized software installed on the server uses computational modules, which include:

- Basic analysis for preliminary testing and verification of the developed software (Static Analysis, Dynamic Analysis, TBvision, TBrun, TBmisra, TBsafe);
- Advanced analysis for preliminary testing and verification of the developed software (Modified Condition/Decision Coverage, Information Flow Analysis, Dynamic Data Flow Coverage, Extract Semantic Analysis);

- Complementary analysis for preliminary testing and verification of the developed software (Test Vector Generation, TBeXtreme, TBmanager, Support for Target Testing, Tool Qualification);
- Additional testing software, which provides ongoing support for the process of developing individual software components.

The presented specialized software, integrated with the computer system supporting avionics software development process management, was used to test the software developed for a helmet-mounted flight-parameter display system.

### 3.3. Technical Implementation of a Computer System Supporting the Management Process

The technical implementation of the computer system supporting the avionics software development management process utilized specialized modular computers (Figure 4), operating as servers and built into an IT cabinet.

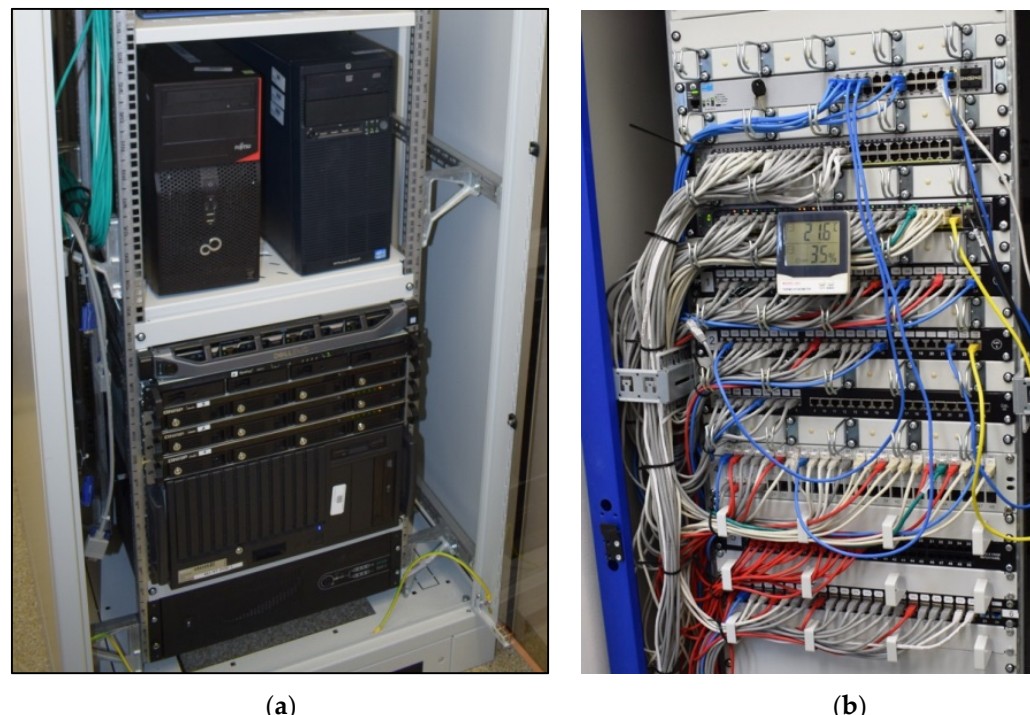

(**a**)                                    (**b**)

**Figure 4.** View of components comprising a computer system supporting the management of flight-safety-critical avionics software development; (**a**) View of the computer system main server; (**b**) View of elements integrated with the IT network.

The AFIT IT network also includes power supply modules, switches, connection panels, cabling, and additional elements (including thermometers and hygrometers to monitor the condition of the air in the server room area). A modular structure of the servers and the application of connection panels (Figure 4b) enables the selection of an IT network configuration that is optimal for the system administrator and users.

The additional equipment of the constructed computer system consisted of a launch stand for the helmet-mounted flight-parameter display system that enables the testing of the developed software. The construction test stand is called a technology demonstrator for a computer system supporting the avionics software development management process.

### 4. Discussion

An example of a system utilizing avionics software that is critical to flight safety is, a helmet-mounted flight-parameter display system.

The helmet-mounted flight-parameter display system is designed to display pilot and navigational information in front of the eyes of Pilot 1 (crew commander) and Pilot 2

(navigator). The system displays information as graphic symbols or in digital form. It enables monitoring flight parameters while simultaneously observing the helicopter's surroundings, without the need to look at the instrument panel when flying. This is particularly important during low-altitude maneuvering flights (e.g., in the mountains), during both day and night, combined with NVG (night vision goggles), which supplement the helmet-mounted system (Figure 5).

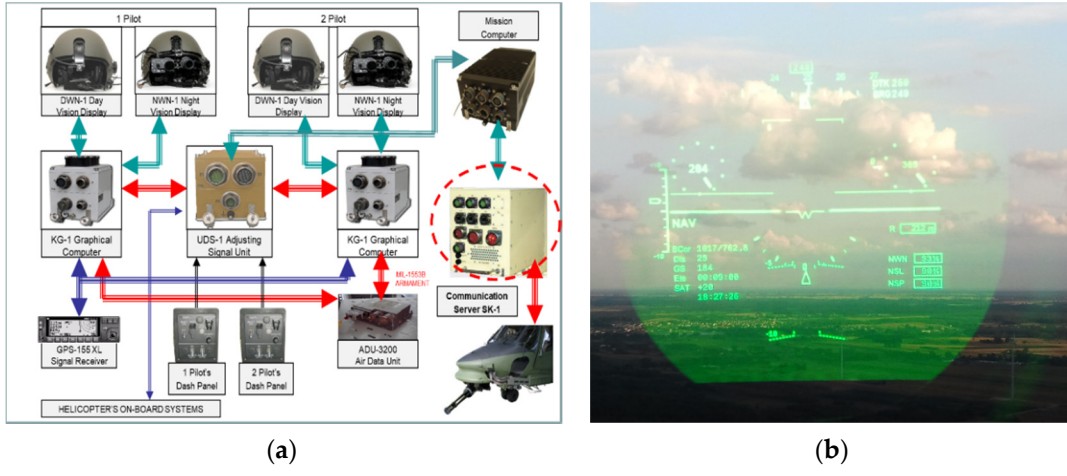

| (a) | (b) |

**Figure 5.** Helmet-mounted flight-parameter display system with avionics software critical to flight safety; (**a**) System's structural diagram; (**b**) daytime helmet display.

The system is fitted with a feature to display pilot and navigational information, as well as to monitor the propulsion system in the form of 16 flight parameters selected on four different display boards and 27 warnings about dangerous situations on board the helicopter (WARN) and helicopter onboard system failures (FAIL) (Figure 6). The advantage of a helmet-mounted flight-parameter display system is the automatic preflight diagnostics and the possibility of entering data into the system.

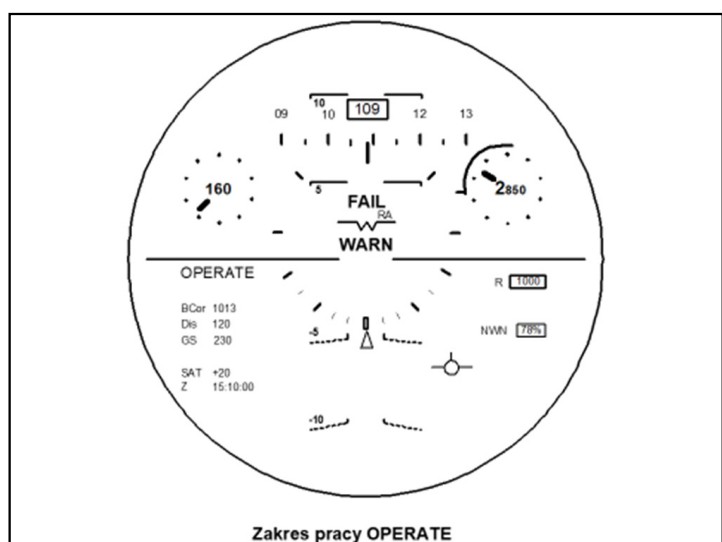

**Figure 6.** Helmet-mounted flight-parameter display system with avionics software critical to flight safety, Warnings regarding dangerous situations (WARN) and Helicopter onboard system failure alerts (FAIL).

As a part of the helmet-mounted flight-parameter display system's lifecycle, the system requirements are derived directly from the system's operating requirements and other conditions associated with flight safety and reliability, as well as operating requirements.

Flight safety requirements result from assessing the security level, which contains functional, integration, and reliability requirements for a given system. Requirements on the error level are defined in the course of the security assessment process in order to guarantee system integrity through specifying the system protections and responses in the event of such errors. These requirements are defined for software and hardware with the aim of eliminating or limiting error effects as well as ensuring error detection, tolerance, removal, and avoidance. System processes responsible for improving and assigning system requirements to hardware and/or software lead to the development of the appropriate architecture for the helmet-mounted flight parameter display system.

The software structure of the helmet-mounted flight-parameter display system, after decomposition, covers such elements as graphic computer software and signal matching system software. System software structure is understood as a combination of software configuration and software architecture. Software structure configuration is understood as a set of directories, subdirectories, and files organized in the form of a directory tree. Software architecture is understood as a set of software modules, the interconnections between them, and the external environment these modules work with. A software module is understood as a set of computer files within an organized structure that executes, a specific functional task.

BIOS configuration is understood as the process of setting the BIOS (Binary Input/Output System) parameters available to a programmer, which leads to improved cooperation between the hardware and software layers.

The "Software documentation for the SWPL-1 helmet-mounted display system" and the "Quality Plan for an IT project involving software for the SWPL-1 system" were developed as a base for the construction of the helmet-mounted flight-parameter display system. These documents attempt to satisfy the requirements set out in DO-178C. Quality plan development was preceded by a detailed review of all requirements regarding the product and contract, with particular attention to new and nonstandard requirements according to AQAP 2210, cl. 2.2.2 and AQAP 2105, cl. 3.2.1.

The software, the nonstandard behavior of which, as indicated in the security assessment process, can lead or contribute to a significant error, resulting in conditions limiting the helicopter's functionalities or an additional burden for the pilot, is subject to a special analysis. If a given component is awarded such a level, it will not be approved by the certification body.

### 4.1. Meeting the Software Planning Requirements

The planning process involving the software for the helmet-mounted flight-parameter display system is defined in a way that the requirements are met and the confidence level is adequate to the adopted software's assurance level. The "IT project of SWPL-1 system software" was developed for the purposes of implementing this project. Based on the aforementioned documents and the requirements set out in the "Initial tactical and technical specifications of the flight parameter imaging system for a Mi-17 helicopter" document, it satisfies the requirements of standard DO-178C. The basis for developing this part of the design is, Procedure No. P115, Software Quality Assurance in accordance with AQAP 2210.

The objective of the IT project was to develop software for a helmet-mounted flight-parameter display system, as per the requirements determined by the ordering party. The software, as the outcome of the IT project, ensures the correct functioning of the helmet-mounted flight-parameter display system throughout its entire lifecycle.

AFIT, as the software developer, has additionally prepared a series of requirements associated with the software development process. These requirements are included in the following documents:

- Internal study: "Preliminary requirements regarding the SWPL-1 flight parameter display system software";
- Internal study: "Detailed requirements regarding the SWPL-1 flight parameter display system software".

The software structure of the SWPL-1 flight-parameter display system was described in the document "Software documentation for the SWPL-1 helmet-mounted flight parameter display system". Recognized software engineering methods, supporting programs, resources, and procedures were applied in the course of the IT project's implementation.

### 4.2. Meeting the Software Development Requirements

According to the requirements of standard DO-178C, software development processes for a helmet-mounted flight-parameter display system is contained in the software planning process and the software development process. Processes associated with software development include software requirement definition processes, software design processes, processes associated with software coding, and integration-related processes.

### 4.3. Meeting the Software Integration Requirements

The integration process of the software for a helmet-mounted flight-parameter display system involves software integration and hardware/software integration. Integration processes can be executed when planned transition requirements are met. Integration process inputs are software architecture from software design processes and the source code from software coding processes, whereas the integration process outputs are the object code files with compilation. Integration processes are complete when their objectives and the goals of associated integral processes are met. The object code should be generated from the source code and then compiled.

All files with data parameters should be generated, and software should be integrated into the main computer, target device emulator, or the target device. The software should be implemented in the target computer for the purposes of hardware/software integration. Inappropriate or erroneous inputs detected during the integration process should be forwarded to software requirement processes, software design processes, coding processes, or software planning processes as feedback that requires verification.

### 4.4. Selected Test Results Involving Software Developed for a Helmet-Mounted Flight-Data Display System

The graphic computer of the helmet-mounted flight-parameter display system conveys flight parameter information on the DWN-1 daytime helmet-mounted display or the NWN-1 night-time helmet-mounted display. The information received from the signal matching system, the GPS satellite navigation receiver, and the ADU aerodynamic data unit is presented as graphic symbols or in digital form.

The graphic computer software structure comprises the following elements:

- BIOS processor direct operation system configuration;
- WINDOWS XP Embedded operating system configuration;
- Graphic computer user software configuration;
- Graphic computer user software architecture (Mi17sys main module, Mi17konfigurator configuration module, and Mi17hud display module).

Graphic computer software is embedded in the permanent memory of the CPU motherboard. The calibration results for individual measurement channels are saved in the computer's external memory on the FLASH packet.

Graphic computer software identification contains such information as software name, software ID number, software version ID, software component modules, and the granted license.

The system software is tested in order to demonstrate that it satisfies the basic requirements set out in AQAP 2210 and DO-178C and to demonstrate, with a high degree of confidence, that the errors, which can lead to the unacceptable failure conditions defined in the ARP 4761 security assessment process, have been removed.

The objective of graphic computer software testing is to demonstrate that the object executable code (software code implemented directly in the device) satisfies high-level requirements and is closely related to low-level requirements.

Three test types were distinguished for the graphic computer software:

- Hardware/software integration testing in order to verify the correct functioning of software on the target computer;
- Software integration testing in order to verify the relationships between software and component requirements and verify the implementation of software requirements and their components within the software architecture;
- Low-level testing in order to verify the implementation of low-level requirements.

If the test cases and the corresponding test procedures that were developed in order to test the hardware/software integration or software integration satisfy the basic coverage and structural coverage requirements, there is no need to duplicate low-level testing. Replacing equivalent low-level tests with high-level tests can lower testing efficiency due to the reduced number of tested functionalities.

Such an example is a "software module" implementing configuration and calibration functions called the "Mi17Konfigurator", which was embedded in a specific hardware environment; this is significantly different from a standard configuration of an IT system. This is why special "human–machine" communication methods were used.

The "Mi17Konfigurator" module, implemented on the graphic computer, closely cooperates with other software modules. The input devices that the software cooperates with are the "*BRIGHTNESS*" manipulator-knob and the "*OPERATING MODE*" button.

As the software system implementing configuration and calibration functions within the helmet-mounted flight-parameter display system was embedded in a specific hardware environment, which is significantly different from standard IT system configuration, satisfying the requirements of user-friendly communication methods was ensured.

The created graphical interface met the requirements of user-communication using, a very limited set of "inputs–outputs". The application of a window-based interface enabled the intuitive operation of the software.

The output device for a military aircraft pilot is the DWN-1 daytime helmet-mounted display. The software recognizes it as a monochrome computer monitor. The "*BRIGHTNESS*" manipulator is used as a device to select software system options, and the "*OPERATING MODE*" button acts as a device confirming the selection. Manipulation results are displayed on the helmet-mounted display on an ongoing basis. The graphical interface system consists of cooperating graphic elements called widgets. Widget examples include graphical buttons, labels, check-boxes, and numerical fields.

The reliability of pilot and navigational information presented on the SWPL-1 helmet-mounted flight-parameter display system consists of three levels:

- Level I is the test of system readiness and efficiency (reporting and alerting malfunctions); integrity is the system's ability to notify the user about the fitness for use within the navigation process in a timely manner;
- Level II is the correction of system errors and scaling; a data-sensitive (faulty measurement data) sent to a pilot or system user as a result of processing individual datasets within this system;
- Level III is matching the indications and pilot supports; reliability is the measure of the pilot's confidence in the correctness of information indicated on board an aircraft.

In order to ensure the required reliability of pilot and navigational information presented on the helmet-mounted flight-parameter display system, the developed software was subject to complex tests utilizing a constructed computer system supporting the avionics software development process.

Selected results of conducted tests involving a fragment of the developed software for a helmet-mounted flight-parameter display system are shown in the figures below.

Figure 7 shows the types and number of errors detected in the software by the MISRA

C:2012 module within the static analysis [32–38].

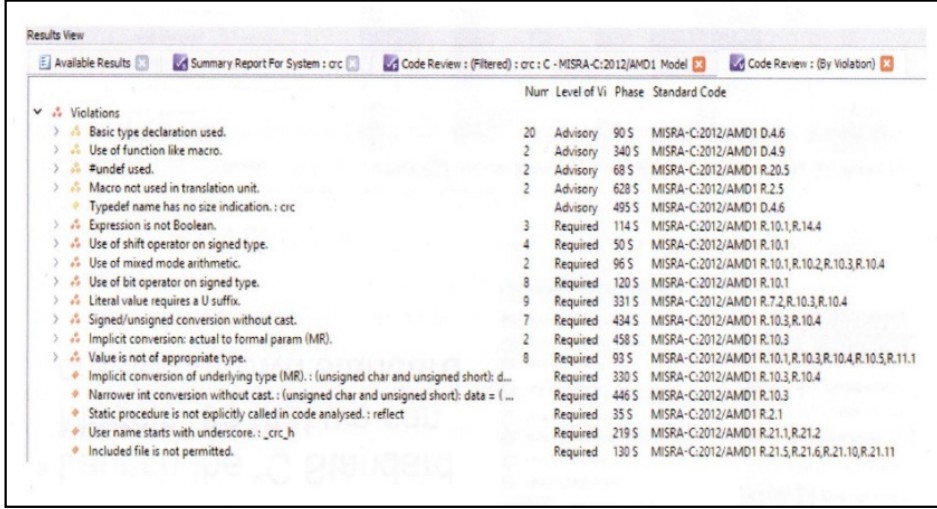

**Figure 7.** Software fragment test results for the static analysis by the MISRA C:2012 module, showing the types and number of errors within the tested software.

Figure 8 shows a graph with the number of erroneous function calls detected in the software code by the MISRA C:2012 module within the static analysis [39–41].

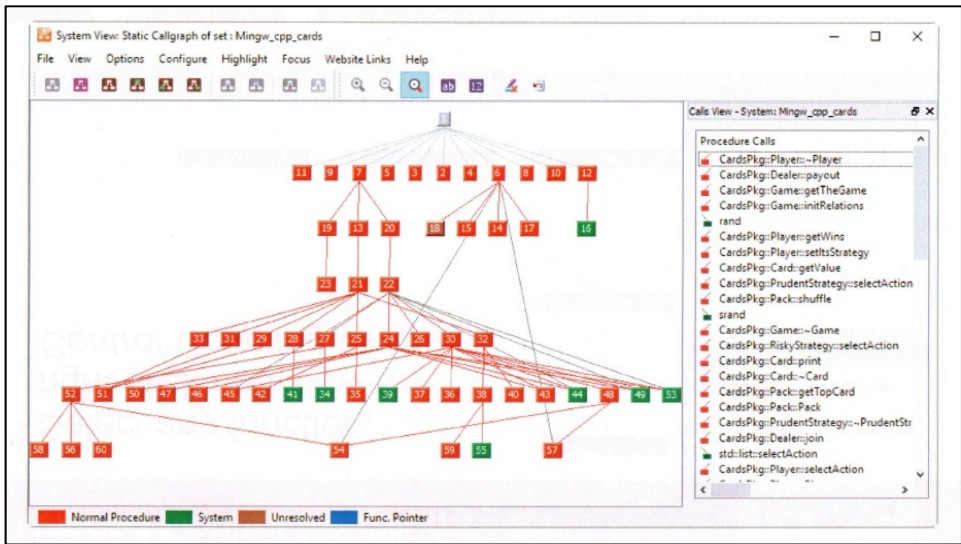

**Figure 8.** Software fragment test results for the static analysis by the MISRA C:2012 module, showing the number of erroneous function calls detected within the tested software.

Figure 9 shows a software quality assessment in terms of satisfying selected requirements, implemented by the MISRA C:2012 module within the static analysis.

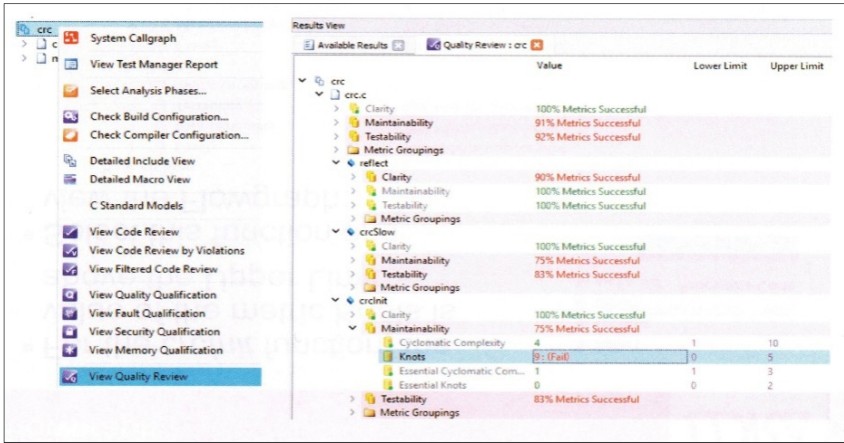

**Figure 9.** Software fragment test results for the static analysis by the MISRA C:2012 module, showing the percentage quality assessment of the tested software.

The conducted verifications showed the presence of numerous errors within individual sections of the software, which enabled their elimination in the target version of the software developed for a helmet-mounted flight-parameter display system [42].

## 5. Conclusions

The progressive computerization of a modern aircraft induces the fitting of onboard equipment with appropriate software, which should satisfy the requirements of flight safety and reliability. Software failures in onboard electronic devices can lead to enormous casualties and material losses. This is why developing such software requires its designers to use particular methods and procedures that are aimed at minimizing errors.

Currently, widely used electronic onboard equipment consists of two fundamental layers, for which appropriate standards have been developed and adopted by international aviation institutions: DO-254 (which sets out requirements in terms of equipment, comprised of various types of electronic circuits, e.g., logic gates, specialized ASIC circuits, PLF programmable logic structures, FPGA) and DO-178C (which sets out requirements in terms of software for various hardware layers, including BIOS, BSP, and operating systems that applications referring to libraries and drivers ).

As the aforementioned layers interact, e.g., through generating interruptions and exchanging data, in order to ensure an appropriate level of reliability and safety of electronic onboard equipment, it is required for their design and manufacturing processes to implement operations in conformity with standards DO-254 and DO-178C. The task of this process is to create an electronic device and software, the operation of which will comply with the adopted assurance levels (A, B, C, D, or E), pursuant to standards ARP 4761 and ARTP 4754A.

Standard DO-178C is a basic document containing issues related to software within the certification of avionic systems and devices. Standard DO-178C is not responsible for the determination of software security guarantees. Security attributes within an IT project, the requirements, and functionality must receive additional, mandatory tasks associated with system security. A certification body requires the DO-178C standard to exhibit comprehensive analysis methods aimed at determining the software level. The software level, also called the project guarantee level or object development assurance level, is determined based on a security assessment and risk analysis, studying the outcomes of a failure within a system. Every bit of software that steers, controls, and monitors critical functions in terms of flight safety should receive the highest level of assurance. The number of achieved objectives depends on the software assurance level.

Standard DO-178C is based on formulating appropriate requirements and their subsequent verification in order to demonstrate meeting them. An advantage of standard DO-178C, a very practically valuable one, is that it does not define the way in which given

requirements are to be satisfied and verified. This means that it enables the use of various tools, methods, and techniques within the software creation process. However, it should be noted that standard DO-178C is a multifaceted process; hence, its implementation at AFIT required a lot of effort and resources. Being aware of the need to satisfy appropriate quality and reliability standards, relative to the developed avionics software, meeting the requirements of standard DO-178C seems obvious and necessary.

The objectivity of verification and validation processes is ensured through their "independence" from the development team. Implementing the requirements of standard DO-178C within the process of developing avionics software dedicated to the SWPL-1 CYKLOP helmet-mounted imaging system, AFIT developed "Documentation of the software for the SWPL-1 system" and "Quality Plan for an IT project involving software for the SWPL-1 system".

The ability to predict vulnerabilities and bugs within the developed software (i.e., the impact of structural damage or hacker attacks) in future software versions can be crucial in assessing the risk associated with avionics software security throughout its lifecycle. Some of the existing quantitative models for analyzing software vulnerabilities were developed based on dedicated software data, which is why actual scenario implementation may be limited. The proposed method for predicting software vulnerabilities based on the use of branching processes supports the avionics software development process, with a minimized impact on flight safety.

The constructed system for computer-aided management of avionics software development, in accordance with standard DO-178C, has been executed based on modular servers and specialist software, with implemented applications testing the developed source code. AFIT, as the source code owner, has the right to modify it by adapting the constructed avionics devices to new functions on board modernized aircraft.

One of the main advantages of the constructed system is the feature of electronic supervision over documents created within a project, both at the software planning stage as well as the software development and integration stage, hence allowing us to obtain the collective information required by the project manager and for audits and inspections.

The system at AFIT is a significant research tool in the computer-aided management of the avionics software development process. It enables the expansion of the features of the current ISO-9001 and AQAP 2210 quality assurance systems existing at AFIT with procedures associated with meeting DO-178C standard requirements.

**Author Contributions:** Conceptualization, M.Z. and A.S.; methodology, A.S.; software, M.Z.; validation, A.S., A.P. and G.K.; formal analysis, M.Z.; investigation, A.P.; resources, A.P.; data curation, G.K.; writing—original draft preparation, M.Z.; writing—review and editing, A.S.; visualization, A.P.; supervision, G.K.; project administration, M.Z.; funding acquisition, A.P. All authors have read and agreed to the published version of the manuscript.

**Funding:** This research received no external funding.

**Institutional Review Board Statement:** Not applicable.

**Informed Consent Statement:** Not applicable.

**Data Availability Statement:** Queries about data availability may be directed to the corresponding author.

**Conflicts of Interest:** The authors declare no conflict of interest.

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
