# Peer review of "Computer Life-Cycle Management System for Avionics Software as a Tool for Supporting the Sustainable Development of Air Transport"

_sustainability, doi:10.3390/su13031547_

Round 1

Reviewer 1 Report

This paper presents a framework to support the management of developing and certifying avionic software that is critical to flight safety. The study is interesting and can be useful. My comments to further improve the paper are as follows:

  • The mathematical derivations on pages 8 and 9 need to be carefully revised. There’re many unreadable symbols, which gave me a hard time understanding the results.
  • Several figures are illegible.

Author Response

Answer 1: Thank you very much for your valuable point of view and we have revised the math results on pages 8 and 9. Please see the revised section which was sent to you.

Reviewer 2 Report

The article is very interesting and its topic is a niche.
There is no purpose of the study in the abstract. This is important as many readers use the information in the executive summary before reading the whole thing. In the introduction, there is also no clearly defined goal of the work. Please complete this.
The material and methods have been described extensively. However, please check the patterns carefully, if there is any error in the spelling.
I do not understand the validity of the reference to the literature e.g. in verse 436-437 (overall results). If these are the results of original research and you don't compare the results with those published by others, why do you do it? Please explain.
I also do not see the validity of the reference to literature, e.g. in lines 679 and 686-687. What is this literature for? Are there such results too? An explanation is needed.
References to the literature in the text of the study should be corrected. The order of citation is incorrect, ie the number 4 comes after the number 5. This should not be the case (lines 44 and 58).
The list of literature should be corrected so that it is adapted to the requirements of the Publishing House (there are two ways of writing here - you start once with the surname, another time with the first name).

Author Response

Thanks a lot for your suggestion and valuable point of view. We added the purpose of the work and changed the summary. Please see the article summary section. We changed references to the literature throughout the text of the article and improved the list of literature adapted to the requirements of the publishing house, we also improved the formulas on pages 8 and 9. Please see the revised section which was sent to you.